# The Role of Outcome Imbalance in Fairness Over Time

## Abstract

We study fairness dynamics in the dropout prediction task and ask whether temporal changes in fairness reflect model bias or shifting outcome imbalance. Using sex as the sensitive attribute (A), we equalize $P(Y \mid A)$ and keep it constant over time through stratified resampling, comparing four scenarios that vary between original and matched data in the training and test sets. Equalizing group–label dependence substantially reduces fairness gaps, yet relative fairness instability persists, suggesting that temporal fairness variation can arise even in the absence of group–outcome imbalance.

## 1 Introduction and Related Work

Machine learning models are rarely deployed in static environments. The data they encounter evolves, and the relationship between sensitive attributes and target outcomes often shifts over time. For instance, the probability of college dropout may differ across demographic groups and vary across time periods used for training the model (e.g., predicting dropout after one semester or after one year of study) [1]. Recent work on fairness under distribution shift highlights how changes in either group proportions or conditional outcomes can lead to fairness drift even for fixed models [3, 2]. Therefore, in this work, we quantify fairness dynamics over time by isolating the effect of changing group–label dependencies.

## 2 Data and Methods

We use nationwide administrative data on student enrollments in Danish education, previously analyzed for dropout prediction and fairness dynamics [1], covering multiple time points during students' enrollment when predictions are made. Each observation represents a student's enrollment at a given time point, described by demographic, institutional, and temporal features, and labeled by eventual dropout. The sensitive attribute (A) is *sex* (binary, as recorded by Statistics Denmark) and is included as a predictor. The prediction target is dropout. The dataset spans multiple academic periods, allowing us to examine temporal dynamics in fairness. For each period, a predictive model (logistic regression) is trained and evaluated on temporally separated data. This setup captures the evolution of fairness metrics as both the population characteristics and the relationship between the target and sensitive attribute change over time. The proportion of enrollments for female and male students and their dropout rate differences are presented in Figure 3. To isolate the effect of group–label dependencies, we construct a matched version of the data in which the conditional dropout rate $P(Y \mid A)$ is equalized across groups and constant over time. Matching is done via stratified resampling, preserving group proportions while adjusting the dropout rate to the overall mean of dropout rate for all enrollments in the dataset (31%). This removes systematic differences in outcome prevalence between groups, allowing us to assess whether fairness variation persists when such dependencies are controlled. Using this matching-based approach, we evaluate fairness dynamics across four experimental scenarios: (1) **Original data:** original training and test data,

(2) **Experiment A:** original training and matched test data, (3) **Experiment B:** matched training and original test data, and (4) **Experiment C:** matched training and matched test data. For each scenario, we retrain the model for each period of enrollment and compute standard fairness metrics (group differences in Accuracy score, AUC, Precision, Recall, and True Negative Rate) across time. We quantify fairness instability as the temporal standard deviation of group fairness differences, averaged across all above-mentioned performance metrics and normalized by their mean absolute magnitude to capture fluctuations independently of overall unfairness.

# 3    Results

As shown in Figure 1, model performance differences between females and males varied across experimental settings. In the original data, sex-based disparities were pronounced early in enrollment and fluctuated over time. Matching the test set (**Experiment A**) reduced unfairness for several metrics (e.g., TNR, Recall) and, for some metrics such as Accuracy score and Precision, even reversed the direction of disparities. Matching only the training data while keeping the original test set (**Experiment B**) led to smaller performance differences for most metrics. Finally, matching both the training and test sets (**Experiment C**) produced the smallest performance gaps overall. Figure 2 summarizes these trends quantitatively. The left panel shows that the average magnitude of fairness gaps (*Mean Absolute Fairness Gap*) decreased progressively from the original data to Experiment C. The right panel shows that the relative fairness instability remained roughly constant across experiments (and even slightly rose for experiment C), suggesting that while fairness improved substantially, its proportional stability over time was not reduced.

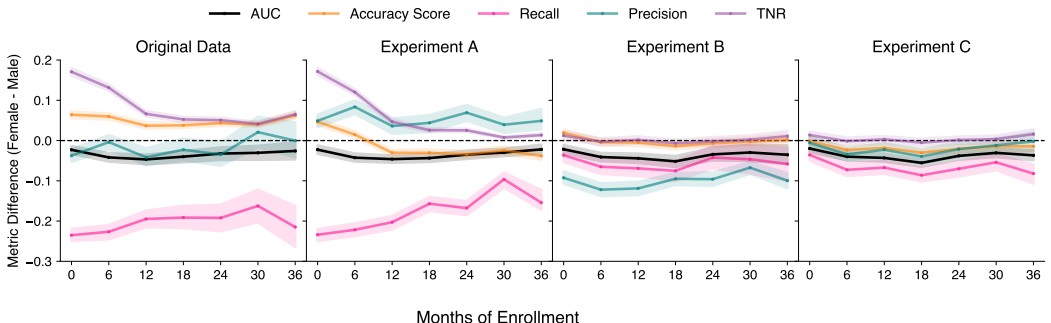

**Figure 1: Model performance differences for sex-based groups across experimental settings.** Shaded regions represent 95% confidence intervals based on bootstrapping.

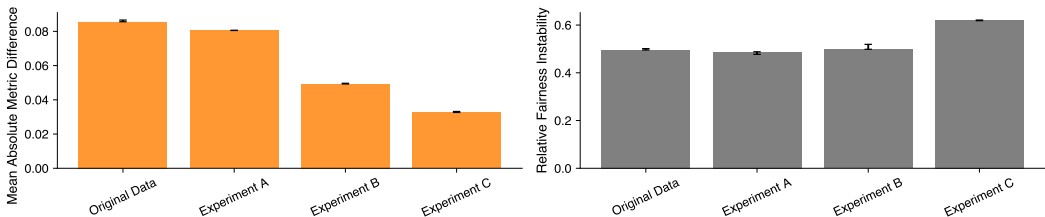

**Figure 2: Mean absolute performance differences for sex-based groups (left) and relative fairness instability (right) across experimental settings.** Error bars indicate 95% confidence intervals estimated from bootstrap resampling.

# 4    Conclusive Discussion

In this tiny paper, we show that much of the unfairness reflected in metric differences arises from target distribution differences across groups, especially in the test set. Yet even with matched outcome distribution in the test set, temporal variability persisted, highlighting the need for fairness research to address both static and dynamic sources of bias in evolving data environments.

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

# A    Appendix

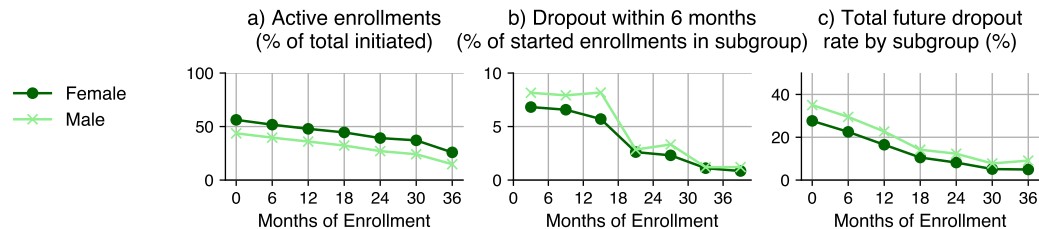

**Figure 3: Enrollment proportions and dropout patterns by sex.** The plots show the proportion of enrollments for female and male students (left), dropout within 6 months (center), and total predicted dropout rates (right) over months of enrollment [1].