# OpenReview forum: "The Role of Outcome Imbalance in Fairness Over Time"
_EurIPS.cc/2025/Workshop/UPLB — UPLB2025_

### Official Review · Reviewer_zC5m · 2025-10-27
**tiny paper - small contribution**

**Rating:** 6
**Confidence:** 4

**Review:**

## overall comment

Although the question of fairness over time and how it may shift, is interesting and relevant, the answer (beginning of) provided by this (tiny) paper is quite limited.

There is:
- an interesting question asked
- it's demonstrated to exist on a given dataset (fairness shoft over time)
- no methodological novelty
- no conceptual novelty
- no **strong** experimental result(s)
- no software
- no proofs


## soundness

This tiny paper seems technically correct, although the reason for performing experiments A and B is unclear to me.
It seems like train and test data should be pre-processed the same way (matched or not matched). Experiment A makes sense in that matched test set is equivalent to a more acute measure (fair measure) of accuracies or other metrics. Usefulness of Experiment B is quite obscure. Experiment C makes perfect sense.

Also, the notion of "fairness instability" is defined with words. An equation, even in appendix, would help clarify.

## relevance to the workshop

This paper is in line with the following themes of the workshop:
- **Dataset shift and out-of-distribution generalization**

And, to some extent:
- **Class and subpopulation imbalance**


## (bonus) significance

limited (see overall comment)

---

### Decision · Program_Chairs · 2025-11-03

Accept (Poster)